

# Identification of human phosphoglycerate mutase 1 (PGAM1) inhibitors using hybrid virtual screening approaches

Numan Yousaf[1,*], Rima D. Alharthy[2,*], Maryam[1], Iqra Kamal[1], Muhammad Saleem[3] and Muhammad Muddassar[1]

[1] Department of Biosciences, COMSATS University Islamabad, Islamabad, Pakistan
[2] Department of Chemistry, Science and Arts College, King Abdulaziz University, Jeddah, Saudi Arabia
[3] School of Biological Sciences, University of the Punjab, Lahore, Pakistan
[*] These authors contributed equally to this work.

## ABSTRACT

PGAM1 plays a critical role in cancer cell metabolism through glycolysis and different biosynthesis pathways to promote cancer. It is generally known as a crucial target for treating pancreatic ductal adenocarcinoma, the deadliest known malignancy worldwide. In recent years different studies have been reported that strived to find inhibitory agents to target PGAM1, however, no validated inhibitor has been reported so far, and only a small number of different inhibitors have been reported with limited potency at the molecular level. Our *in silico* studies aimed to identify potential new PGAM1 inhibitors that could bind at the allosteric sites. At first, shape and feature-based models were generated and optimized by performing receiver operating characteristic (ROC) based enrichment studies. The best query model was then employed for performing shape, color, and electrostatics complementarity-based virtual screening of the ChemDiv database. The top two hundred and thirteen hits with greater than 1.2 TanimotoCombo score were selected and then subjected to structure-based molecular docking studies. The hits yielded better docking scores than reported compounds, were selected for subsequent structural similarity-based clustering analysis to select the best hits from each cluster. Molecular dynamics simulations and binding free energy calculations were performed to validate their plausible binding modes and their binding affinities with the PGAM1 enzyme. The results showed that these compounds were binding in the reported allosteric site of the enzyme and can serve as a good starting point to design better active selective scaffolds against PGAM1enzyme.

## INTRODUCTION

In metabolic pathways of cellular respiration, glycolysis is considered as the main step to produce energy in the form of adenosine triphosphate (ATP). The eighth step of glycolysis is catalyzed by a glycolytic enzyme called phosphoglycerate mutase 1 (PGAM1). PGAM1 reversibly isomerizes 3-phosphoglycerate (3PG) to 2-phosphoglycerate (2PG) through 2, 3-bisphosphoglycerate (2BPG) intermediate in order to proceed aerobic glycolysis (*Hitosugi et al., 2012*; *Liu et al., 2017*). Glycolytic enzyme phosphoglycerate mutase 1 (PGAM1) has

Corresponding author
Muhammad Muddassar,
mmuddassar@comsats.edu.pk

potential stimulating roles in a number of human cancers. In cancer cells, PGAM1 is highly up regulated that cause rapid production of cancer cells and tumor growth (*Xu et al., 2016*; *Zhang et al., 2017*).

Human genome contains two PGAM genes thereby two iso-types of dPGM genes exist, which includes PGAM1 or PGAM-B and PGAM2 or PGAM-M. Their encoded products share high degree of homology with 81% identity, similar length of 254 and 253 amino acids respectively. Moreover, they form homodimers and heterodimer that are functionally identical. The PGAM1 is present in different body parts such as liver, brain, kidney, red blood cells (RBCs), and early fetal skeletal muscle, whereas PGAM2 can be found in adult skeletal muscles and myocardium (cardiac muscles). The MB heterodimer is found in heart and in the late fetal or neonatal muscle (*Xu et al., 2014*).

In humans the active enzyme exists in the form of dimer which is formed by the C strands of two monomers with antiparallel alignment. In the center of the dimer chloride ion is located which strengthens the interactions of the two monomers. Low concentration of chloride ion accelerates dPGM activity. Whereas its higher concentration inhibits dPGM activity, as chloride ion binds at the enzyme's active site which is rich with positively charged residues and compete with the substrate thereby inhibit enzyme's activity (*Xu et al., 2014*).

Currently PGAM1 has only few reported classes of inhibitors that include MJE3, PGMI-004A, and a natural chemical epigallocatechin gallate (EGCG), the major polyphenol derived from *Camellia sinensis* (green tea) (*Evans et al., 2005*) and xanthone derivatives. MJE3 was initially identified as PGAM1 inhibitor but it also inactivates PGAM1 by covalently modifying its K100 residue, because of its spiroepoxide substructure (*Li et al., 2017*). PGMI-004A is a verified PGAM1 inhibitor; it exhibits anticancer properties in mice models carrying human xenograft tumors. These two reported inhibitors have low molecular potency (*Liu et al., 2017*). Although PGAM1 third inhibitor EGCG has 30 times higher molecular potency than PGMI-004A but due to its polyphenol structure and the off-target effect, it cannot be considered as a good inhibitor (*Li et al., 2017*). Recently xanthone derivatives (24 *N*-xanthone benzene sulfonamides) have been designed and synthesized as PGAM1 inhibitors. Most of them exhibit high potency against PGAM1 as compared to PGMI-004A and their anti-proliferation activity was moderate on different cancer cell lines (*Ma et al., 2014*; *Wang et al., 2018b*). Due to few reported PGAM1 inhibitors, there is a need to design and identify better active compounds that can bind to PGAM1 enzyme to block its metabolic activity.

## MATERIALS AND METHODS

### Preparation of active, decoys and database compounds

The reported inhibitors of PGAM1 were selected as active compounds dataset. Decoys dataset was prepared using physicochemical properties of the PGMI-004A co-crystal ligand. The criteria such as HBA<9, HBD<3, molecular weight and logP value ± 463.38 g/mol, ± 4 respectively was used to fetch the decoys [18] from ChEMBL database https://www.ebi.ac.uk/chembl/. Decoy molecules exhibit the similar physicochemical

properties like active molecules; however, they are chemically distant and act as false positive results in validation studies. Chemdiv database was prepared by implementing drug like filter criteria to remove unwanted chemical structures, duplicates and salts. Stereoisomers of molecules were checked and tautomeric states were generated at pH 7.0. OMEGA tool was used to generate one hundred conformers per molecule.

## Query model generation and optimization

Shape and color based query models were built using vROCS tool (*Poongavanam & Kongsted, 2013*) implemented in OpenEye scientific software (https://www.eyesopen.com/) (*Grant, Gallardo & Pickup, 1996*). PGMI-004A co-crystal ligand was used for generating query models which were validated by plotting receiver operating characteristic (ROC) curve (*Poongavanam & Kongsted, 2013*). The ROC curves usually discriminate true positive and false positive compounds in the dataset. In our experiments, the graphs were represented by fraction of the active on $Y$-axis and fraction of the decoy dataset on $X$-axis. Different features were iteratively added and removed to obtain the best query model that can distinguish true actives from inactive compounds during virtual screening.

## ROCS shape and electrostatic potential similarity-based screening

ROCS shape and color-based screening method was used to align the best query model with chemical database compounds on the basis of shape and color matching, the best match was selected based on TanimotoCombo scoring function. Screened hits possessed higher TanimotoCombo score were subjected for electrostatic potential similarity matches using the EON software implemented in OpenEye software. It calculates the partial charges similarity between database and query molecule, and then calculates the ET-Combo score which is the combination of shape-Tanimoto and PB-electrostatic similarity coefficient. Hits having ET-Combo >1 were selected for further screening (*Imran et al., 2020*).

## Protein and ligand structure preparation for molecular docking

Human PGAM1 structure (PDB-ID: 5Y2I) was retrieved from protein data bank (https://www.rcsb.org/) (*Berman, Henrick & Nakamura, 2003*). The crystal structure of PGAM1 in complex with ligand PGMI-004A was in dimeric form consisting of two chains (Chain A and B). Protein preparation wizard embedded in Schrödinger's Maestro tool (*Sastry et al., 2013*) was used for preprocessing of structure. All water molecules and other co-crystal ligands were removed and structure was minimized using OPLS2005 forcefield to remove all steric clashes. Similarly, for molecular docking studies compounds were prepared using Ligprep tool embedded in Schrodinger software (*Guasch et al., 2013*). Different ionization states and tautomeric forms were generated at pH 7.0 (*Li, Robertson & Jensen, 2005*).

## Grid generation and glide docking

Before docking studies, a Grid box was generated around the allosteric site of the PGAM1 structure. The size of the outer box was adjusted up to 17Å and size of inner box was fixed to 12Å, 12Å, and 15Å for $X$, $Y$ and $Z$ axis respectively. The receptor grid generation panel was used to generate the grid at the allosteric sites of the prepared protein. At the end by

using Glide SP docking (precision tool from Schrodinger software) (*Friesner et al., 2004*) prepared ligands were docked within generated grid.

## Identification of toxicological and pharmacokinetics properties

Toxicological and pharmacokinetics studies were conducted on virtually screened compounds mined from Chemdiv database by using online tool PreADMETv.2.0 (https://preadmet.bmdrc.kr/adme/) (*Lee et al., 2004*; *Lee et al., 2003*). ADME data *i.e.,* toxicity, absorption, and distribution were calculated by means of selected following parameters: toxicological in Ames test, carcinogenicity for rats and mouse, absorption in human intestinal absorption, cell permeability, cell permeability Maden Darby canine kidney, in-vitro P-glycoprotein inhibition, permeability in Caco2 cells and distribution in blood–brain barrier, plasma protein binding (*Araújo et al., 2020*).

## Identification of biological activity of screened compounds

Biological activities were predicted in order to test the possibility of the biological effects of the selected compounds. Activity predictions were made using the online PASS server (http://www.pharmaexpert.ru/passonline/) (*Lagunin et al., 2000*) which is the freely available software to predict the biological effects of the compounds based on their chemical structures. PASS server suggests the activity of the compounds based on their chemical formula using multi-level atom neighbors (VMA) descriptors. Activities of compound with following effects had been selected: antiallergic, antiasthmatic, Alzheimer's disease treatment, and antidiabetic.

## Molecular dynamic (MD) simulation method

Molecular dynamic simulation (MD) was carried out using the NAMD software (https://www.ks.uiuc.edu/Research/namd) based on Charm run and molecular dynamic force fields. The parameters of the inhibitors were described by General Amber Force Field (GAFF) and atomic charges were calculated through the Antechamber package of Amber tools 21 (https://ambermd.org/AmberTools.php) (*Macke et al., 2010*). Subsequently, protein structures of PGAM1 in complex with inhibitors were prepared for simulation using the tLeap module of amber tools 21. The ff14SB force field and TIP3P water model were described for solvent-solvent simulation. Protein ligand complexes were immersed in 10Å octahedral water box and then neutralize by adding $Na^+/Cl^-$ ions. After that, we initialize the system for simulation includes: energy minimization of the complexes by removing undesired contacts between non-bonded adjacent atoms then, the temperature of the system is increased gradually from 200k to 250k to 300k in order to stabilize the system (*de Oliveira et al., 2019*; *Jorgensen et al., 1983*). This step is done in three further steps with each steps ran for about 5,000 steps, at each integration step velocities are reassigned and the temperature is increased. In the final step, the simulation was run in the production phase at 310K temperature and 1 atm pressure by using the NPT ensemble, for 50 ns in order to study the behavior of proteins after binding with the ligands.

## Binding free energy calculation

Binding free energies of the protein-ligand complexes were calculated using molecular mechanics Poison Boltzmann calculations MMGBSA free energy algorithm (*Sun et al.,*

*2014*). The $\Delta G_{total}$ was calculated using following equation:

$$\Delta G_{total} = \Delta G_{complex} - [\Delta G_{protein} + \Delta G_{ligand}]$$

$\Delta G_{total}$ is the sum of interaction energy of the gas phase in between protein-ligand complex. However free energy calculation per residue was calculated by following equation:

$$\Delta G_{MMGBSA} = \Delta G_{vdW} + \Delta G_{ele} + \Delta G_{GB} + \Delta G_{Surf}$$

### Residues free energy decomposition

In order to find the key residues involved in the binding of protein and ligand, MMGBSA free energy decomposition program of Amber21 was also applied to the complexes. The interaction of ligand and protein can be described by the following equation:

$$\Delta G_{inhibitor\_residue} = \Delta G_{vdW} + \Delta G_{ele} + \Delta G_{ele,sol} + \Delta G_{nonpol,sol}$$

### Computational alanine scanning

The key residues involved in the protein-ligand binding free energy were mutated to alanine to find their role in the total binding free energies of the complexes. The mutated complexes were subjected to recalculate the binding free energy. The total binding free energy of the two systems (wild type and mutant) can be calculated by the following equation:

$$\Delta\Delta G_{bind} = \Delta G_{bind}^{mutant} - \Delta G_{bind}^{wildtype}$$

The flowchart of overall study is shown in Fig. 1.

## RESULTS AND DISCUSSION

### Molecular docking of reported compound

Different classes of PGAM1 inhibitors have been reported which include xanthone derivatives, anthraquinone derivative, MJE3 and epigallocatechin (EGCG) (*Evans et al., 2005*; *Wang et al., 2018a*). Molecular docking studies were performed to find the sites in PGAM1 protein at which reported compounds are binding. The analyses showed that among all reported classes xanthone derivatives are binding at the allosteric site with reasonably good glide score values. The binding interactions of protein residues such as hydrogen bonding, pi-pi stacking, etc. with xanthone derivatives were observed as reported in co-crystalized structure. The key residues involved in the interactions are Phe22, Arg90, Tyr92 and Arg191 (*Wen et al., 2019*).

### ROCS query model generation and virtual screening

Since the co-crystal ligand 8kx (xanthone derivative) showed good binding score among other reported compounds. Therefore, it was selected for the development of query model. Finest ROCS query models were produced and optimized in four phases, in first phase, all pharmacophore features were applied for model building. In second phase model was generated by reducing the number of pharmacophoric features to 5A and 4R (A: hydrogen bond acceptor, R: ring), phase 3 have 4A and 3R pharmacophoric features and in the last

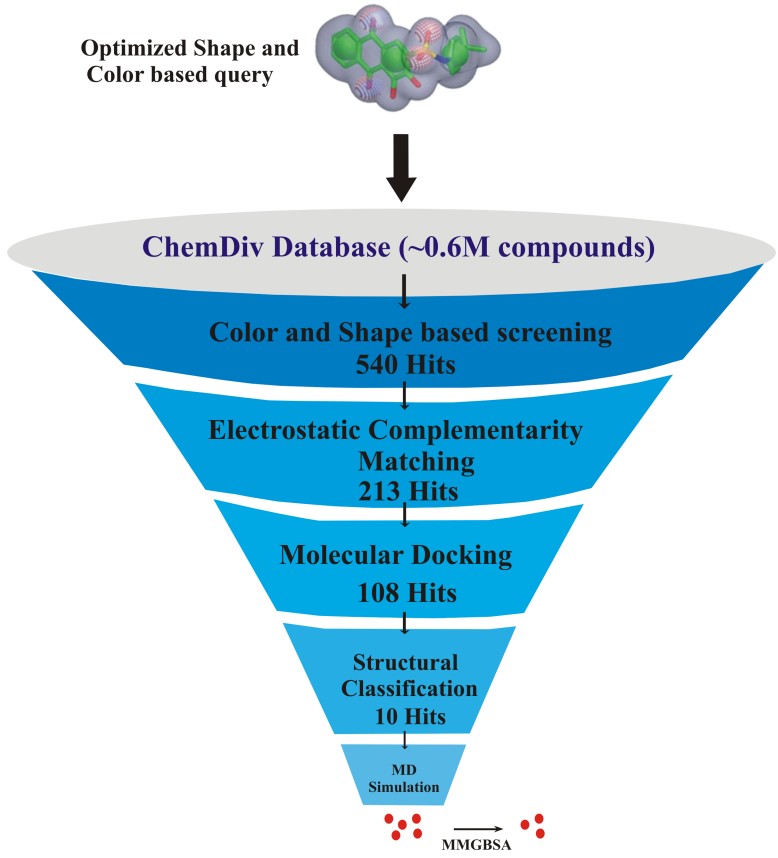

**Figure 1   The flowchart of the study.**

phase query model was built on 3A, 3R features as shown in Fig. 2A. The query models were sequentially validated to observe their ability to discriminate the active compounds from the decoys. The dataset used for the validation contained fifteen active compounds and six hundred decoys retrieved from ChEMBL database. The query-1 showed the poor enrichment with an area under curve (AUC) value of 0.78, whereas query 2 and 3 showed better screening performance and yielded AUC 0.84 and 0.83 respectively. The query 4 discriminated the active and decoys significantly better than other three queries with an AUC value of 0.87. The ROC plots shown in Fig. 2B are representing the fraction of active ligands (the true positives) on the $y$-axis and the fraction of decoys (the false positives) on the $x$-axis.

After generating optimized query model, the Chemdiv database was screened by matching the query features with its compounds. Since the query contains both shape and color features, therefore it was speculated that unknown compounds having similar characteristics with query will show the biological activities (*Kearnes & Pande, 2016*). Shape and color Tanimoto scores were used to rank the overlaying compounds. TanimotoCombo score ranges from 0 to 2 was used to select the hits, which means hits with value closer

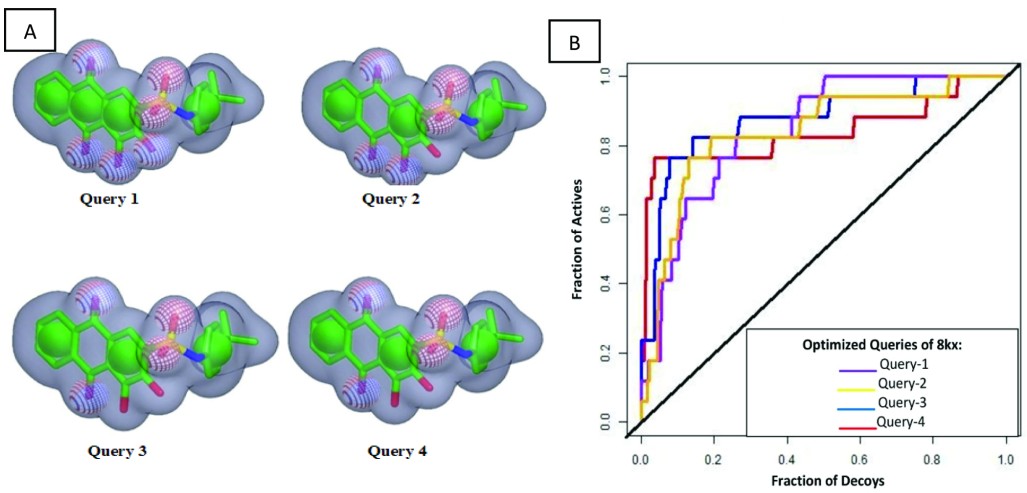

**Figure 2** **Query model optimization.** (A) Multiple ROCS queries on co-crystal ligand. The green color is indicating hydrophobic regions *i.e.*, four aromatic rings, four red regions are showing acceptors and a single blue region is representing donor. (B) Validation of multiple query model of co-crystal ligand 8kx represented by AUC curve.

to 2 will exactly overlay its shape and features to query model, while the value closer to 0 indicates shape and chemical-features are not matching (*Kearnes & Pande, 2016*; *Li et al., 2018*). Using this scoring scheme, 540 compounds were obtained with TanimotoCombo score ranges from 1.20−1.53. Then these hits were subjected to compare their partial charges complementarity with cocrystal ligand using PB Electrostatic Tanimoto and shape Tanimoto (ET_Combo score) implemented in EON (*Rush et al., 2005*).

## Molecular docking based virtual screening and clustering

To further narrow down the size of initially screened hits from ROCS and EON based approaches, hits were docked at the allosteric site of PGAM-1 target protein. All the hits were sorted based on the glide score and the compounds having glide score comparative to the binding affinity of co-crystal ligand were selected for their structural clustering analysis. The purpose of structure based clustering was to classify the compounds into groups based on structure similarity (*Mangiatordi et al., 2017*). Based on the glide scores and structural diversity, five compounds (C1 = 6144-0309, C2 = E470-1348, C3 = C301-8900, C4 = F540-0157 and C5 = L464-0403) were selected from each cluster. Among the top five compounds, C2 exhibit the good binding pose having highest glide score of −6.902 Fig. 3. On comparison, docking score of virtually screened top hits compounds were relatively equal to the selected ligands of PGAM1. Selected compounds also indicated the greater binding interaction within the allosteric site residues. Compound C1 formed one hydrogen bond and salt bridge with Lys100 and pi interaction with Phe22. The oxygen of sulfonamide group of C2 was involved in two hydrogen bonds with Arg90, Arg191 while NH group formed one hydrogen bond with Tyr92. Additionally, a pi interaction was observed with Phe22. Similarly, Arg191 and Tyr92 were making hydrogen bonds with sulfonamide oxygen

**C1**

**T.C**= 1.4520

**G. S**= -5.570

**C2**

**T.C**= 1.5430

**G. S**= -6.902

**C3**

**T.C**= 1.3580

**G. S**= -6.054

**C4**

**T.C**= 1.2730

**G. S**= -6.446

**C5**

**T.C**= 1.4330

**G. S**= -6.128

**Figure 3** **The best virtual screen hits.** C1 = 6144–0309, C2 = E470–1348, C3 = C301–8900, C4 = F540–0157, C5 = L464–0403 from Chemdiv database with their chemical structures, ROCS TanimotoCombo scores (TC) and glide docking scores (GS).

and NH group of compounds C3, while Phe22 was involved in pi-pi stacking. In complex of PGAM1-C4, Phe22 was involved in two types of interactions. It made a hydrogen bond with carbonyl oxygen and pi-pi stacking with the benzene ring. The C5 compounds was also making the hydrogen bonds with Tyr92 and Arg191 and pi-pi stacking with Phe22 as shown in Fig. 4A. All these interactions of the screened compounds demonstrated that these were bound to the reported key residues of the allosteric pocket of PGMA1 enzyme (*Wen et al., 2019*). The binding poses of these compounds were also analyzed by comparing with the binding pose of the co-crystal ligand 8kx, which showed that these compounds have similar plausible binding modes as the reference ligand as shown in Fig. 4B.

## Analysis of toxicological and pharmacokinetic properties

Toxicological and pharmacokinetics properties of the selected hits were assessed using online server PreADMET (https://preadmet.bmdrc.kr/adme/). First the absorption levels of the compounds were predicted using different parameters like HIA (%), CaCo-2 (nm/sec), PMDCK (nm/sec), P-gp inhibition. All the compounds showed high value of HIA>94%, which is believed the most important ADME properties. CaCo-2 permeability also showed significant results, value greater than two showed easy absorption of the

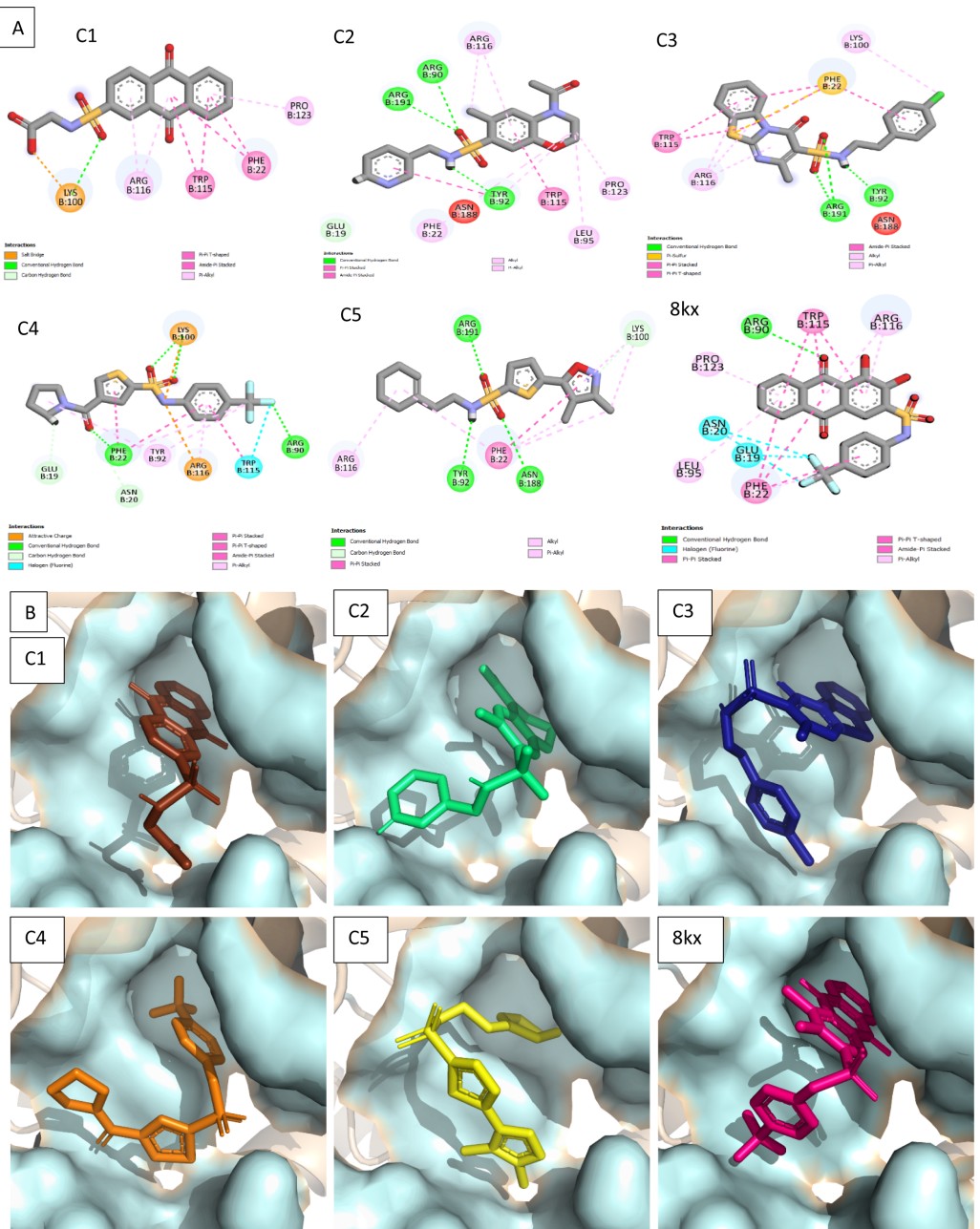

**Figure 4** **Docked poses of hit compounds.** Protein-ligand complexes (C1, C2, C3, C4, C5 and 8kx) with best screened compounds and co-crystal ligand in the binding site. (A) 2D interactions of hits with the allosteric site residues. (B) The binding poses of the best selected hits in the allosteric site with reference to cocrystal ligand 8kx.

compound. Compound C1 showed highest 22.2815 nm/s absorption rate in all reported inhibitors. Moreover, $P_{MDCK}$ permeability value of C1 has also seen substantially increased 204.401nm/s and C4 have moderate and ideal value of 4.13987nm/s. Other important drug

**Table 1  ADME properties calculation of virtually screened compounds using PreADMET (https://preadmet.bmdrc.kr/adme/).**

| Compound | Human intestinal absorption | Cell permeability | Cell permeability maden darby canine kidney | *In-vitro* p-glycoprotein inhibition | Permeability of blood-brain barrier |
|---|---|---|---|---|---|
| | HIA (%) | CaCo-2 (nm/sec) | $P_{MDCK}$ (nm/sec) | P-gp inhibition | BBB (%) |
| C1 | 100.000 | 22.281 | 204.401 | Non | 1.499 |
| C2 | 96.912 | 14.758 | 2.171 | Non | 0.154 |
| C3 | 97.039 | 19.580 | 1.510 | Non | 0.181 |
| C4 | 97.063 | 4.271 | 4.139 | Inhibitor | 0.736 |
| C5 | 97.276 | 11.180 | 0.667 | Non | 1.564 |

absorption barrier in the cell is P-glycoprotein (P-gp). Data showed that it assists in the intestinal penetration of compounds except C4 which dropped the efflux method due to its passive permeability. The Distribution parameter of BBB% was referred to characterize the distribution of the compounds in various tissues *in vivo*. Compounds have BBB% >0.3 can cross the blood–brain membrane easily (*Clark, 1999*). According to data, all compounds were predicted to be good to cross the blood–brain barrier. It was observed the compounds C2, C3, and C4 have showed BBB (%) values less than one which means that these compounds have no activity on central nervous system (CNS) shown in Table 1.

Data in Table 2 illustrates that the values of possibility of biological activity of compounds are higher as compare to possibility of inactive (Pa >Pi) generally in term of antineoplastic, antiallergic and antidiabetic responses. The selected compounds showed Pa>Pi values, indicating the relation with reported biological activities. C1 possess best value for proteasome ATPase inhibitor (Pa = 0.784) and insulysin inhibitor (Pa = 0.637), C2 (Pa = 0.863) have good activity value as omptin inhibitor, C3 (Pa = 0.338) shows better inhibition against antineoplastic, C4 (Pa = 0.808) as an anti-obesity and C5 (Pa = 0.734) for nootropic disease. All the candidate compounds have good biological activity values for different diseases and can be performed as potential inhibitors against different targets.

## MD simulation studies

Molecular dynamic studies of best identified compounds were performed to evaluate the binding modes and stability of selected compounds with PGAM1. MD-simulations studies help to understand the dynamic movement of the protein ligand complex and to calculate their binding free energy. The best fitted compounds (C1 = 6144-0309, C2 = E470-1348, C3 = C301-8900, C4 = F540-0157, C5 = L464-0403) in complex with PGAM1 were subjected to MD simulation for 50 ns time scale. Trajectory analysis were performed by calculating RMSD of back bone atoms, RMSF, and the radius of gyration of all the five compounds in complex with PGAM1. Figures 5, 6 and 7 showed that all compounds (except C4) in simulation system were stable, compact and well fitted in the allosteric site. The protein backbone RMSD helps to analyze the protein-ligand complex stability over the

**Table 2** Pharmacokinetics properties and biological activities of top five virtually screened compounds (C1, C2, C3, C4, and C5), as determined by web server PASS ( http://www.pharmaexpert.ru/passonline/).

| COMPOUND | Possibility of activity ($p_a$) | Possibility of inactive($p_i$) | Biological activities |
|---|---|---|---|
| C1 | 0.150 | 0.141 | Antipruritic, non-allergic |
|  | 0.162 | 0.153 | Antibacterial |
|  | 0.784 | 0.006 | Proteasome ATPase inhibitor |
|  | 0.637 | 0.018 | Insulysin inhibitor |
| C2 | 0.411 | 0.049 | Antiallergic |
|  | 0.863 | 0.004 | Omptin inhibitor |
|  | 0.253 | 0.093 | Alzheimer's disease treatment |
|  | 0.142 | 0.131 | Antineoplastic (renal cancer) |
| C3 | 0.243 | 0.196 | Insulysin inhibitor |
|  | 0.338 | 0.016 | Antineoplastic |
|  | 0.090 | 0.073 | Lipoxygenase inhibitor |
|  | 0.140 | 0.010 | Sulfonylureas |
| C4 | 0.808 | 0.005 | Ant obesity |
|  | 0.650 | 0.009 | Antidiabetic |
|  | 0.290 | 0.070 | Alzheimer's disease treatment |
|  | 0.258 | 0.138 | Antineoplastic (Multi myeloma) |
| C5 | 0.231 | 0.021 | Osteoarthritis treatment |
|  | 0.379 | 0.049 | Antidiabetic |
|  | 0.734 | 0.031 | Nootropic |
|  | 0.169 | 0.162 | Diabetic nephropathy treatment |

simulation time (*Sargsyan, Grauffel & Lim, (2017)*). The RMSD values of the complexes range from 0 Å 3 Å shown in Fig. 5. It can be observed that all complexes showed stability as the RMSD of all complexes remained in the range of ~2.5 to 3 Å after being equilibrated at 5 ns. A small deviation in the RMSD value of PGAM1-C1 complex was observed during 35 to 40 ns but it gained stability towards the end of simulation. The overall behavior of all complexes showed that protein was remained stable during the simulation. Further, the different snapshots of protein-ligand complexes (Fig. 6) were extracted from 0, 5, 10, 15, 20, 25, 30, 35, 40, 45, and 50 ns and aligned to detect the position of the docked compounds and analyze the mobile regions of the protein. The aligned snapshots depicted that C4 compound did not remain at docked site (Fig. 6D) while the other compounds were tightly bound to the allosteric site. The elasticity and fluctuations in protein amino acid residues are calculated by measuring the root mean square fluctuations. The higher RMSF value shows that the protein is more flexible and vice versa (*Martínez, 2015*). In RMSF analysis, the C and N terminal and the loop regions gave higher fluctuation in all five complexes as shown in Fig. 7. The protein residues did not show major fluctuations during simulation however residues range from 100 to 150, showed some minor fluctuations. This region contained the binding sites of protein as well as some loop regions. Besides these, no significant fluctuations were observed at the ligand binding site of PGAM1. Overall PGAM1 in complex with the ligands found stable and no abrupt deviations have been observed during whole simulation. Similarly, no uncertain behavior of the complexes was
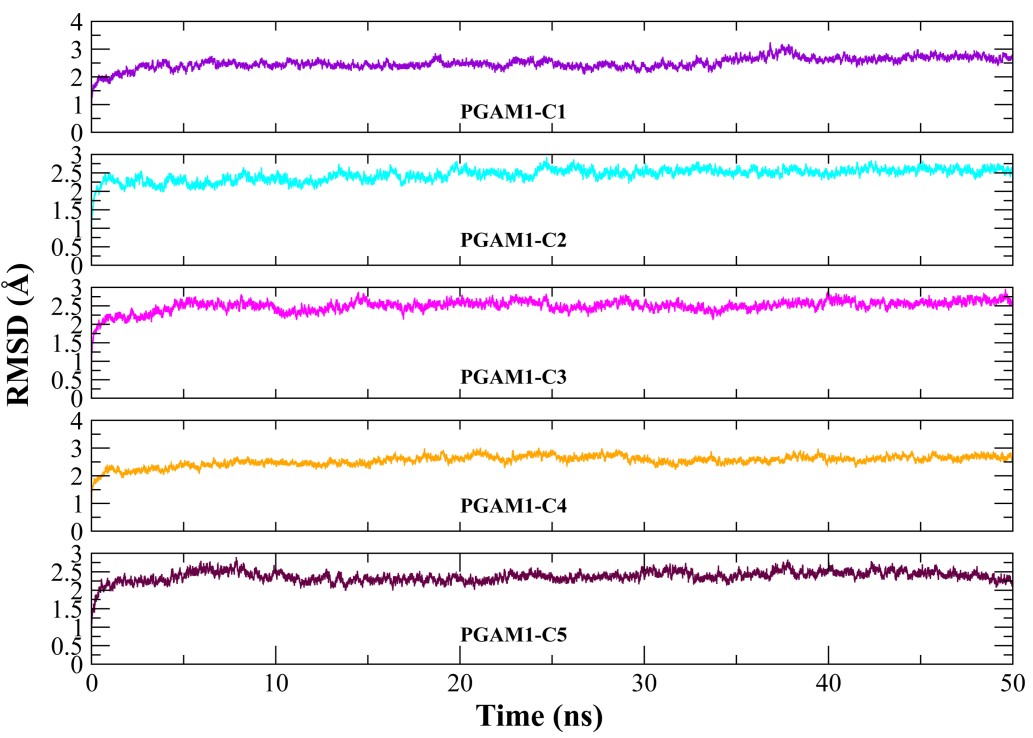

**Figure 5** **The RMSD(Å) *vs.* time(ns) plots.** PGAM-1 enzyme backbone atoms in complex with C1 (red), C2 (blue), C3 (green), C4 (orange), and C5 (pink) hits.

seen during the 50 ns MD simulation which indicated that the virtual hit compounds were well fitted at allosteric site of PGAM1 enzyme.

## Binding free energy studies

Molecular mechanics generalized born surface area (MMGBSA) method was used to calculate the total binding free energy ($\Delta G_{total}$) for all five complexes. $\Delta G_{total}$ value is usually used to estimate the stability of protein-ligand complex. The lower values of $\Delta G_{total}$ indicates that the complex is more stable (*Wang et al., 2021*) and vice versa. It was computed as a sum of protein-ligand complex and the difference of PGAM1 protein and its ligands free energies. The total binding free energy estimated using MMGBSA model is the outcome of the contribution of various protein-ligand interactions such as van der Waals energy (VDWAALS), electrostatic energy (EEL), EGB (electrostatic contribution to solvation free energy by Generalized Born).

The $\Delta G_{VDW}$ of C2, C3 and C5 complexes were found to be $-37.42 \pm 0.26$, $-42.36 \pm 0.19$ and $-35.96 \pm 0.32$ kcal/mol respectively and contributed more in binding affinities. C1 and C4 had limited contribution of $-29.61 \pm 0.17$ and $-17.79 \pm 0.35$ respectively. In the case of $\Delta G_{ELEC}$ the energy component was $-21.66 \pm 0.57$ in C1, $-19.64 \pm 0.29$ in C2, $-21.39 \pm 0.34$ in C3, $-13.24 \pm 0.72$ in C4 and $-15.17 \pm 0.50$ in C5. $\Delta G_{ELEC}$ energy contribution of C1 is highest among all other complexes. After those surface energies were analyzed, C3 complex exhibited highest surface energy binding affinities of $-5.18 \pm 0.01$

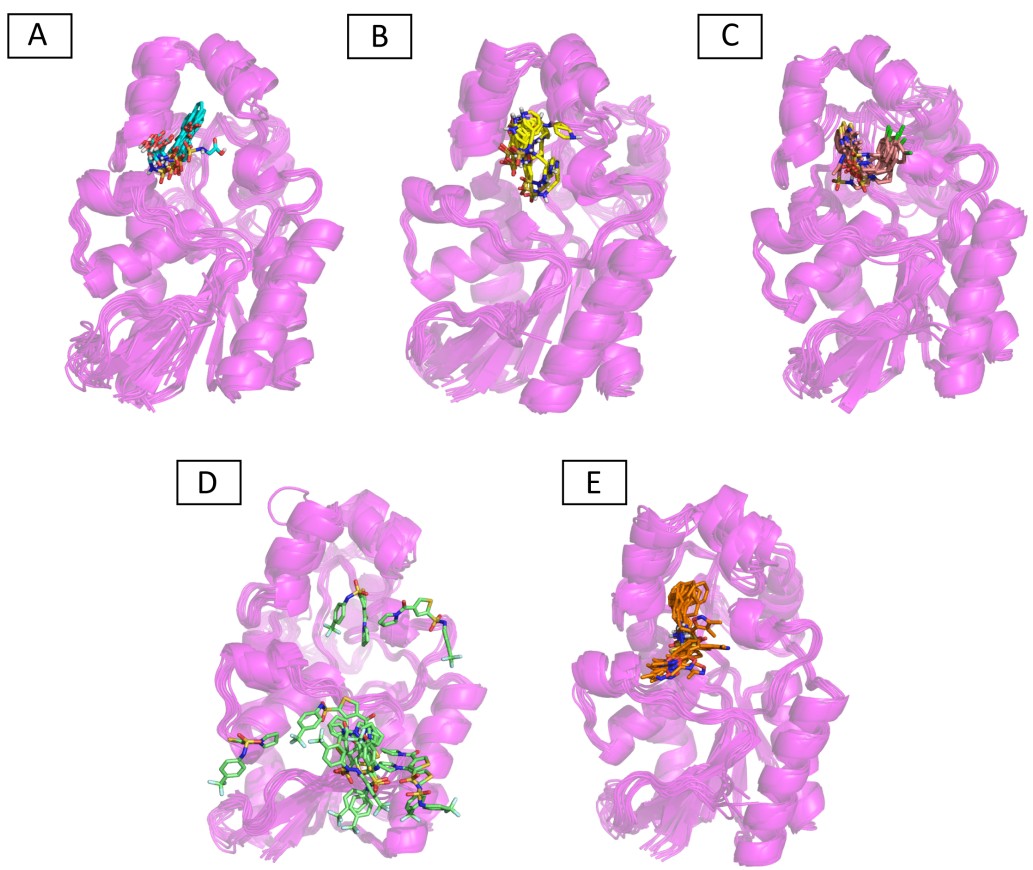

**Figure 6** **The superimposed snapshots of the MD trajectories of C1 to C5 complexes.** (A) PGAM1- C1 (B) PGAM1-C2 (C) PGAM1-C3 (D) PGAM1-C4 (E) PGAM1-C5.

while C4 showed lowest energy. The energy contribution of each energy function of all complexes is shown in Table 3.

The $\Delta G_{total}$ of C2 ($-37.11 \pm 0.36$), C3 ($-41.53 \pm 0.39$) and C5 ($-36.79 \pm 0.62$) were quite better and similar compared to C1 and C4 hit. The differences in the binding energies were due to the difference in the contribution of electrostatic, van der Waal, and surface energies in the protein-ligand complexes. In order to observe the major energy contribution and favorable interactions of residues with selected hits, the total binding free energy was decomposed for each residue of the complexes (*Chohan et al., 2016*), shown in Fig. 8. The decomposition analysis showed that hits were interacting with various PGAM1 residues. The residues like Phe22, Tyr92, Arg90, Lys100, Arg116, Asn188, and Arg191 showed favorable contribution in total binding free energy of all complexes except for C4 complex where Asn20 and Phe22 contribution was very minor. However, Glu19 and Glu89 residue showed unfavorable interactions with C2, C3, and C5 hits, as these interactions had positive binding free energy values. The interactions of C2, C3, and C5 compounds with the PGAM1 allosteric site residues before and after MD simulations are shown in Fig. 9.
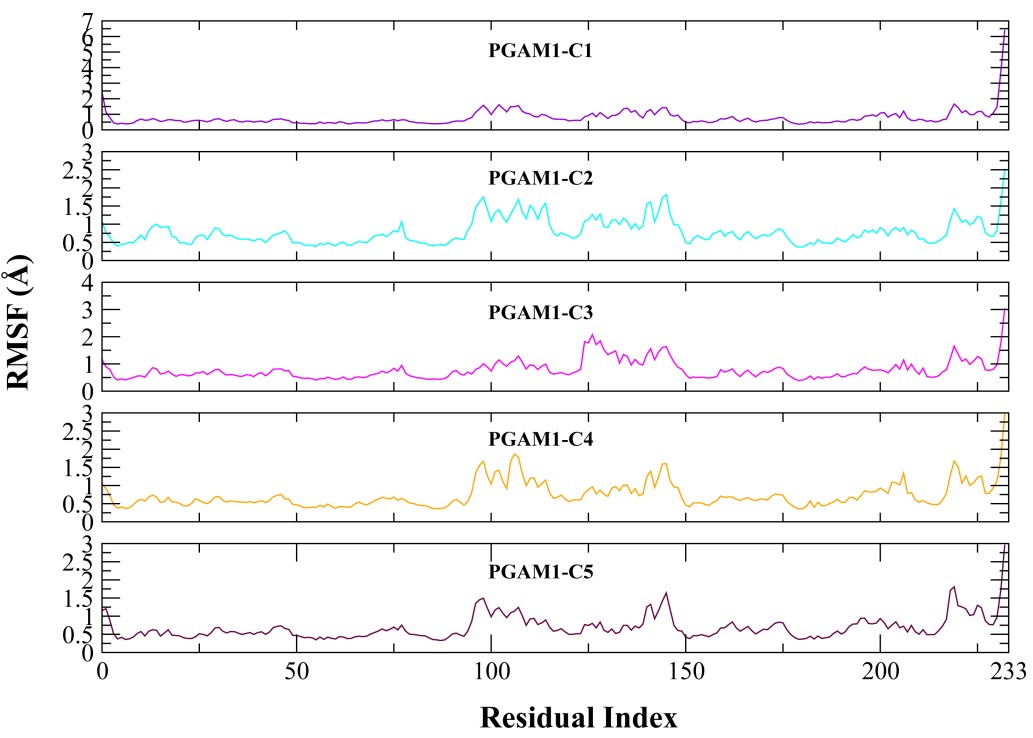

**Figure 7** **The RMSF (Å) plots of complexes.** The RMSF (Å) plots of residues of PGAM-1 enzyme in complex with C1 (red), C2 (blue), C3 (green), C4 (orange), and C5 (pink) hits.

**Table 3** **Binding free energy (kJ/mol) calculation of complex C1, C2, C3, C4, and C5 determine by MMGBSA.**

| Interactions | PGAM1-C1 | PGAM1-C2 | PGAM1-C3 | PGAM1-C4 | PGAM1-C5 |
|---|---|---|---|---|---|
| $\Delta G_{vdW}$ | $-29.61 \pm 0.17$ | $-37.42 \pm 0.26$ | $-42.36 \pm 0.19$ | $-17.79 \pm 0.35$ | $-35.96 \pm 0.32$ |
| $\Delta G_{ele}$ | $-21.66 \pm 0.57$ | $-19.64 \pm 0.29$ | $-21.39 \pm 0.34$ | $-13.24 \pm 0.72$ | $-15.17 \pm 0.50$ |
| $\Delta G_{GB}$ | $38.31 \pm 0.49$ | $25.04 \pm 0.29$ | $27.41 \pm 0.25$ | $25.98 \pm 0.71$ | $19.33 \pm 0.28$ |
| $\Delta G_{surf}$ | $-4.09 \pm 0.02$ | $-5.09 \pm 0.02$ | $-5.18 \pm 0.01$ | $-2.16 \pm 0.04$ | $-4.99 \pm 0.04$ |
| $\Delta G_{gas}$ | $-51.27 \pm 0.55$ | $-57.06 \pm 0.32$ | $-63.76 \pm 0.33$ | $-31.04 \pm 0.55$ | $-51.13 \pm 0.74$ |
| $\Delta G_{solv}$ | $34.21 \pm 0.48$ | $19.94 \pm 0.29$ | $22.22 \pm 0.25$ | $23.81 \pm 0.73$ | $14.34 \pm 0.26$ |
| $\Delta G_{total}$ | $-17.05 \pm 0.23$ | $-37.11 \pm 0.36$ | $-41.53 \pm 0.39$ | $-7.22 \pm 0.34$ | $-36.79 \pm 0.62$ |

## Distance and hydrogen bond analysis

The distances of hit molecules with key residues (Phe22, Tyr92, Arg191) that majorly contributed to the binding free energy, was measured in all trajectories. It was observed that the nitrogen (N1) of Borazine group of C2 was away from the amino group hydrogen (H) of Phe22 in first 20 ns but after that both maintained an average distance of ~4 Å till the end of simulation. Similarly, the distance between sulfur atom (S1) of thiophene ring of C3 and benzene hydrogen (H4) of Phe22 was in the range of ~4−4.5 Å throughout the simulation while the distance between Isoxazolidine ring of C5 and benzene ring of

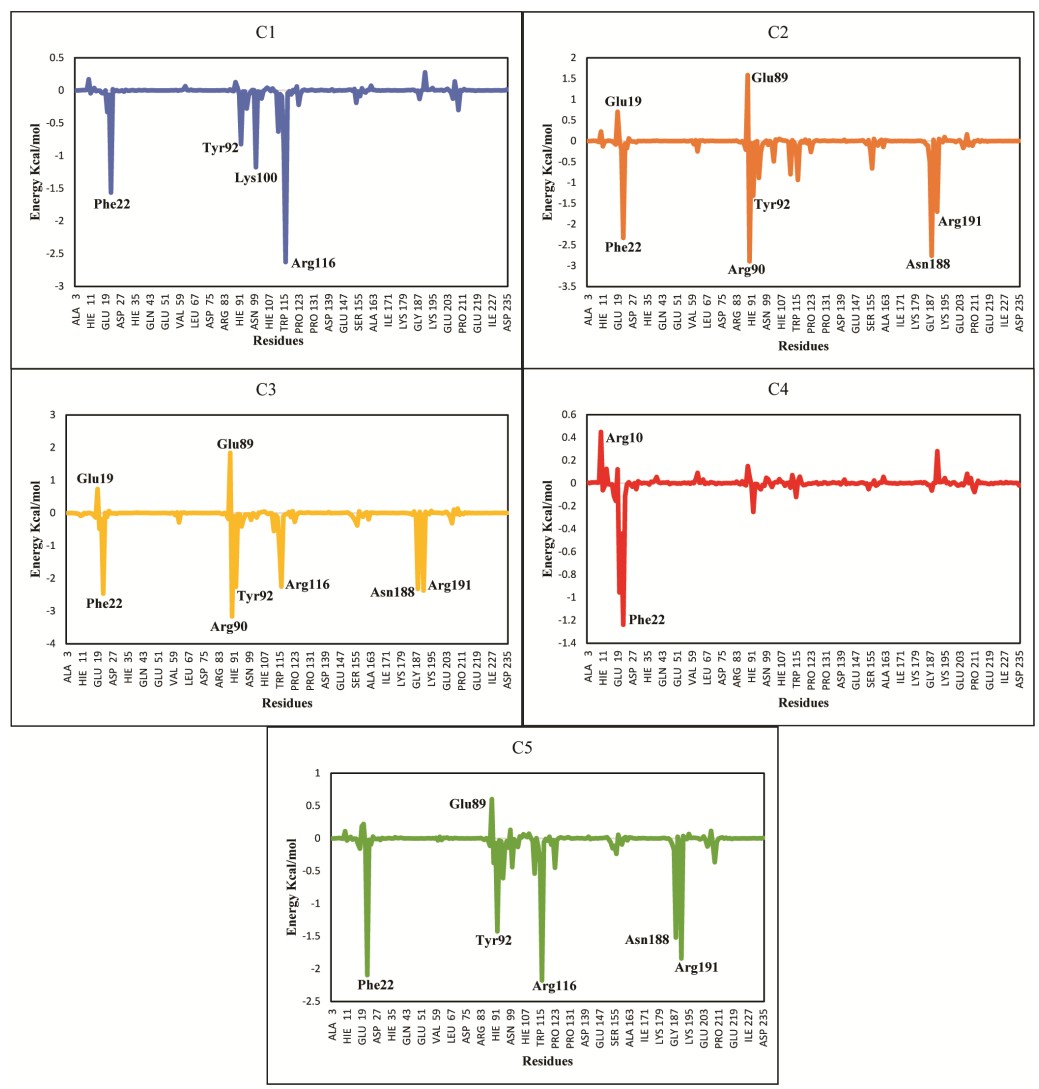

**Figure 8** **Total binding free energy decomposition of all five hits-PGAM-1 complexes.** Key residues with major contribution (Phe22, Arg90, Tyr92, Arg116, and Arg191) are shown in peaks.

Phe22 was ~3.5–6 Å for whole simulation time. The distance between amine group (N2) of C2 and hydroxyl group (OH) of Tyr92 was less than 4 Å throughout the simulation. Similarly, the distance between amine group (N3) of C3 and hydroxyl group (OH) of Tyr92 was ~2−2.5 Å all the time except for some frames where the distance increased to ~5 Å. The distance between the amine group (N2) of C5 and hydroxyl group (OH) of Tyr92 was less than 2 Å till 20 ns and then increased to 4 Å at 22 ns. The distance remained in this range until 49 ns. The distance increased to ~6 Å towards the end of simulation. Moreover, the distance between sulfonamide group oxygen (O3) of C2 and amine group (NH2) of Arg191 was <4 Å till 18 ns and exceed to 10 Å after 20 ns for some time and then attained an average distance of ~7 Å till 50 ns except for small deviations at 43 to

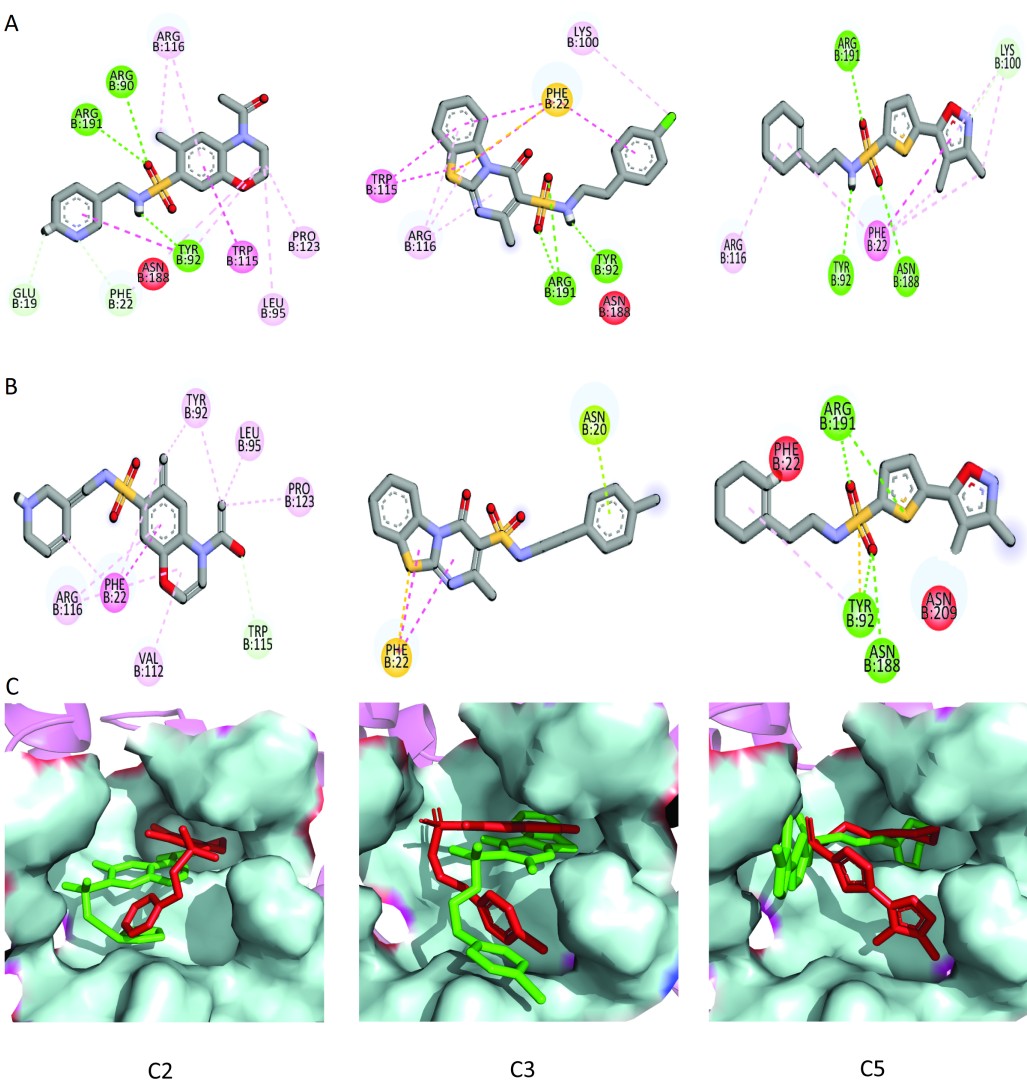

C2                                    C3                                    C5

**Figure 9  The interactions patterns of finally selected hits before and after MD simulations.** (A) The interactions showed at 0 ns (first frame). (B) The interactions at 50 ns (last frame). (C) The binding modes of hits before and after MD simulations. The red sticks are showing poses (first frame) while the green sticks are indicating the poses in last frame.

45 ns. The similar trend was also observed in the distance between sulfonamide group oxygen (O3) of C3 and amine group (NH1) of Arg191. However, the distance between sulfonamide group oxygen (O1) of C5 and amine group (NH1) of Arg191 was in the range of ~2–5 Å  throughout the simulation (Fig. 10). Further, hydrogen bond analysis was performed to observe the hydrogen bond patterns between the protein binding pocket and selected ligands. Hydrogen bond formation or distortion is a principal factor during simulation so large number of hydrogen bonds show stronger affinity between the ligand and protein (*Surti et al., 2020*). The hydrogen bonding plots of the three complexes are shown in Fig. 11. The average number of hydrogen bonds between C2 and protein in first

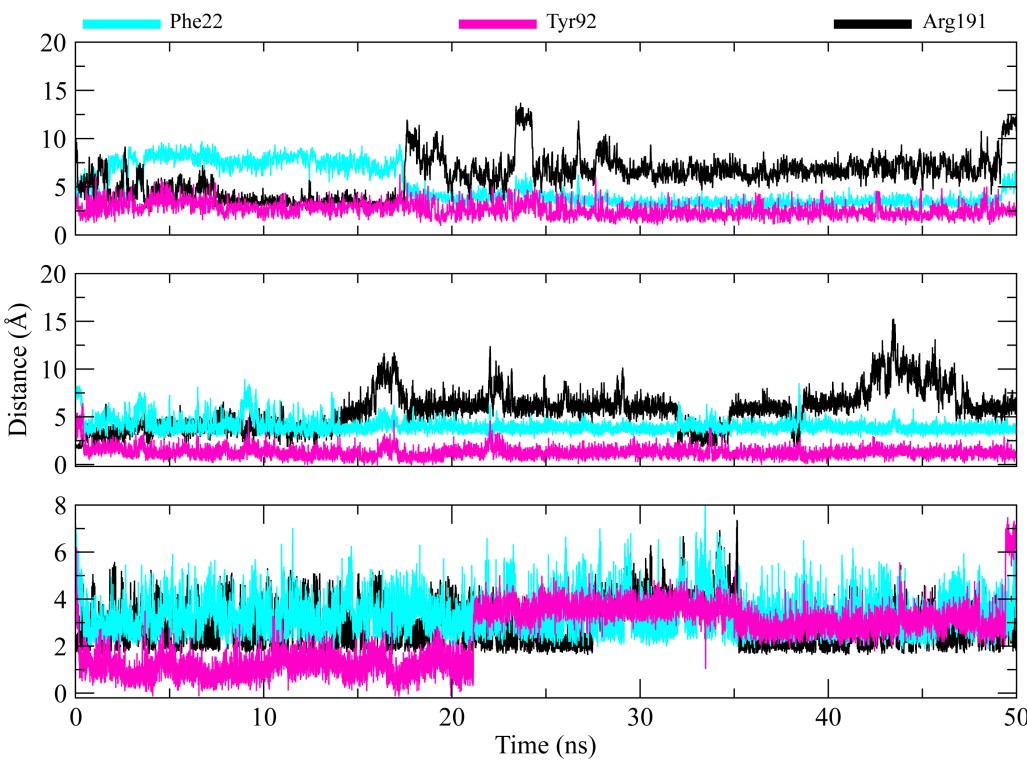

**Figure 10   Distance plots among key residues and selected hits.** Distance between the key residues (Phe22, Tyr92, Arg191) and selected hit compound throughout the MD simulation. Cyan plots indicate the distance between Phe22 and hit compounds, Magenta shows the Tyr92-Hits distances,while black plots show the distances between hits and Arg191.

25 ns were 1.60 while in the second half, the average number of hydrogen bonds were 1.18. Similarly, the average number of hydrogen bonds between C3 and protein were 1.45. Among the three hits, the average number of hydrogen bonds of C5-PGAM1 were higher than other two hits with a value of 2.21. The hydrogen bonding analysis showed that the selected hits made strong and stable complexes with protein.

## Alanine scanning of selected complexes

The three complexes (C2, C3 and C5) showed maximum binding free energies were further investigated to identify the key role of Phe22, Tyr92, and Arg191 residues in the total binding free energy of protein-ligand complexes using computational alanine scanning. This method helps to calculate the energy contribution of individual residue in total binding free energy of protein-ligand complex (*Liu et al., 2018*). One speculation of the computational alanine scanning is the effect of mutation on the structural perturbation of system is so small that the binding free energy of mutated system can be obtained from the wildtype trajectory. So, three complexes were selected based on the better binding free energies and the contribution of Phe22, Tyr92, and Arg191 in total binding free energy was estimated one by one. The change in the binding free energy of three complexes and

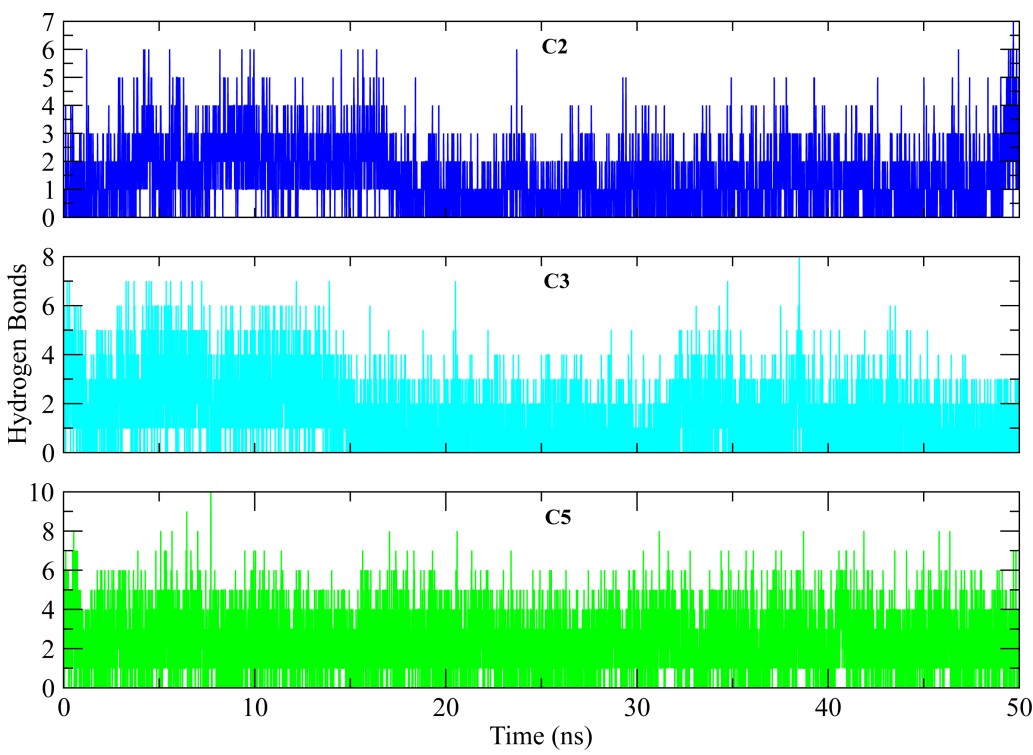

**Figure 11** **Hydrogen bond analysis of hits with key residues.** The number of hydrogen bonds formed between the binding site of protein and selected hits (C2, C3, C5) during simulation. The plots show the consistent hydrogen bonding among all complexes.

entropy changes can be observed in Table 4. In PGAM1-C2 complex, the mutation of Arg191 to alanine played a significant role in the loss of binding free energy as total energy reduced to $-30.70 \pm 0.29$ with an energy loss of $-6.41 \pm 2.52$ as compared to the other two mutations where energy loss of Tyr92 and Phe22 mutations was $-3.74 \pm 1.40$ and $-4.49 \pm 1.08$ respectively. Similarly, the contribution of Arg191 was higher in PGAM1-C3 complex as this mutation lead to an energy loss of $-6.67 \pm 4.26$ that was higher than $-5.56 \pm 1.49$, and $-4.49 \pm 1.44$ for mutations of Tyr92 and Phe22 respectively. In case of PGAM1-C5 complex, same trend was observed as Arg191 mutation caused a loss of $-5.72 \pm 2.90$ energy while the energy loss for Phe22 and Tyr92 mutation caused an energy loss of $-5.19 \pm 1.72$ and $-4.41 \pm 2.58$ respectively. The major difference in binding free energies was due to the mutation of Arg191 which indicates that Arg191 have critical role in binding with the ligands.

**Table 4 Binding free energy (kcal/mol) calculation of mutant complexes C2, C3 and C5.**

| Interactions | PGAM1-C2 R191A | PGAM1-C2 Y92A | PGAM1-C2 F22A | PGAM1-C3 R191A | PGAM1-C3 Y92A | PGAM1-C3 F22A | PGAM1-C5 R191A | PGAM1-C5 Y92A | PGAM1-C5 F22A |
|---|---|---|---|---|---|---|---|---|---|
| $\Delta G_{vdW}$ | $-35.62 \pm 0.27$ | $-33.90 \pm 0.23$ | $-33.50 \pm 0.24$ | $-41.48 \pm 0.18$ | $-38.57 \pm 0.17$ | $-38.53 \pm 0.17$ | $-35.22 \pm 0.33$ | $-33.16 \pm 0.30$ | $-32.08 \pm 0.28$ |
| $\Delta G_{ele}$ | $-11.10 \pm 0.29$ | $-19.35 \pm 0.29$ | $-19.11 \pm 0.28$ | $-8.63 \pm 0.29$ | $-19.58 \pm 0.34$ | $-20.77 \pm 0.35$ | $-4.13 \pm 0.29$ | $-12.60 \pm 0.44$ | $-14.51 \pm 0.51$ |
| $\Delta G_{GB}$ | $20.83 \pm 0.28$ | $25.11 \pm 0.30$ | $24.63 \pm 0.28$ | $20.22 \pm 0.20$ | $27.04 \pm 0.23$ | $26.99 \pm 0.24$ | $13.04 \pm 0.20$ | $17.98 \pm 0.29$ | $19.41 \pm 0.31$ |
| $\Delta G_{surf}$ | $-4.81 \pm 0.02$ | $-4.82 \pm 0.02$ | $-4.64 \pm 0.02$ | $-4.96 \pm 0.01$ | $-4.83 \pm 0.01$ | $-4.72 \pm 0.01$ | $-4.75 \pm 0.04$ | $-4.60 \pm 0.03$ | $-4.40 \pm 0.04$ |
| $\Delta G_{gas}$ | $-46.72 \pm 0.28$ | $-53.65 \pm 0.31$ | $-52.61 \pm 0.31$ | $-50.11 \pm 0.34$ | $-58.15 \pm 0.35$ | $-59.31 \pm 0.34$ | $-39.35 \pm 0.53$ | $-45.76 \pm 0.67$ | $-46.60 \pm 0.71$ |
| $\Delta G_{solv}$ | $16.01 \pm 0.28$ | $20.28 \pm 0.30$ | $19.98 \pm 0.28$ | $15.25 \pm 0.20$ | $22.21 \pm 0.23$ | $22.27 \pm 0.24$ | $8.28 \pm 0.21$ | $13.38 \pm 0.27$ | $15.00 \pm 0.29$ |
| $T \Delta S$ | $-22.62 \pm 0.84$ | $-22.62 \pm 0.84$ | $-22.62 \pm 0.84$ | $-20.23 \pm 1.26$ | $-20.23 \pm 1.26$ | $-20.23 \pm 1.26$ | $-24.62 \pm 1.15$ | $-24.62 \pm 1.15$ | $-24.62 \pm 1.15$ |
| $W \Delta G_{total}$ | $-37.11 \pm 0.36$ | $-37.11 \pm 0.36$ | $-37.11 \pm 0.36$ | $-41.53 \pm 0.39$ | $-41.53 \pm 0.39$ | $-41.53 \pm 0.39$ | $-36.79 \pm 0.62$ | $-36.79 \pm 0.62$ | $-36.79 \pm 0.62$ |
| $M \Delta G_{total}$ | $-30.70 \pm 0.29$ | $-33.37 \pm 0.34$ | $-32.62 \pm 0.34$ | $-34.86 \pm 0.25$ | $-35.94 \pm 0.41$ | $-37.03 \pm 0.40$ | $-31.06 \pm 0.56$ | $-32.37 \pm 0.55$ | $-31.59 \pm 0.57$ |
| $\Delta\Delta G_{bind}$ | $-6.41 \pm 2.52$ | $-3.74 \pm 1.40$ | $-4.49 \pm 1.08$ | $-6.67 \pm 4.26$ | $-5.58 \pm 1.49$ | $-4.49 \pm 1.44$ | $-5.72 \pm 2.90$ | $-4.41 \pm 2.58$ | $-5.19 \pm 1.72$ |

**Notes.**

$T \Delta S$, The entropy changes; $W \Delta G_{total}$, Binding free energy of Wild type complex; $M \Delta G_{total}$, Binding free energy of Mutant complex; $\Delta\Delta G_{bind}$, Difference in mutant and wild type energy values.

## CONCLUSIONS

In this study the optimized shape and structure-based approaches yielded the five hit compounds having the sulfonamide as linking group to attach different chemical fragments. Molecular docking studies reveal that among finally selected hits, C3 binds at PGAM1 allosteric site better than other screened virtual hits. Further molecular dynamics simulations and binding free energies calculations suggest that C2, C3 and C5 hits are tightly binding at the allosteric binding site of PGAM1 enzyme with $-37.11 \pm 0.3$, $-41.53 \pm 0.39$ and $-36.79 \pm 0.62$ Kcal/mol binding free energies respectively with no structural distortions throughout the simulations time. Computational alanine scanning suggests that Phe22, Tyr92 and Arg191 residues are playing important role in binding interactions of these hits with PGAM1 enzyme, which is in agreement with the experimental findings (*Wen et al., 2019a*). The predicted ADMET properties also suggest that these compounds can be used as lead compound after performing the biological assays for further optimization.

### Funding

The Higher Education Commission of Pakistan provided the funds vide project No. 8094/Balochistan/NRPU/R&D/HEC/2017 to purchase computational resources. The funders had no role in study design, data collection and analysis, decision to publish, or preparation of the manuscript.

### Grant Disclosures

The following grant information was disclosed by the authors:
Higher Education Commission of Pakistan: 8094/Balochistan/NRPU/R&D/HEC/2017.

### Competing Interests

The authors declare there are no competing interests.

### Author Contributions

- Numan Yousaf conceived and designed the experiments, performed the experiments, analyzed the data, prepared figures and/or tables, authored or reviewed drafts of the article, and approved the final draft.
- Rima D. Alharthy conceived and designed the experiments, performed the experiments, analyzed the data, prepared figures and/or tables, authored or reviewed drafts of the article, and approved the final draft.
- Maryam conceived and designed the experiments, performed the experiments, analyzed the data, prepared figures and/or tables, authored or reviewed drafts of the article, and approved the final draft.
- Iqra Kamal conceived and designed the experiments, performed the experiments, analyzed the data, authored or reviewed drafts of the article, and approved the final draft.

_____________________________________________

- Muhammad Saleem performed the experiments, analyzed the data, prepared figures and/or tables, authored or reviewed drafts of the article, and approved the final draft.
- Muhammad Muddassar conceived and designed the experiments, analyzed the data, prepared figures and/or tables, authored or reviewed drafts of the article, provided software support, and approved the final draft.

### Data Availability

The data is available at Figshare: Muddassar, Muhammad (2022): Trajectory files. figshare. Dataset. https://doi.org/10.6084/m9.figshare.21069949.v1.

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
