# Peer review of "Identification of human phosphoglycerate mutase 1 (PGAM1) inhibitors using hybrid virtual screening approaches"

_PeerJ, doi:10.7717/peerj.14936_

## Round 0.1 · original submission · Major Revisions

In too many recent publications, and also in this submission, description of the MD results is quite poor and routine: authors simply show evolution of global RMSD, RMSF, radius of gyration and number of hydrogen bonds between ligand and protein. This is quite insufficient, and often useless: for example, radius of gyration is a relatively featureless quantity that mostly can only be used to detect dramatic structural changes akin to denaturation or unraveling of major parts of the structure. RMSD is somewhat better than Radius of gyration, but it is also too often presented without any kind of critical analysis. Importantly, it depends on whether the protein contains a single domain or several domains connected by hinge regions: one could (e.g.) get a large-ish RMSD (which is often a sign that something has gone wrong) in a system where nothing untoward has happened simply if the hinge region makes the relative orientation of each domain change slightly. It is, therefore, better (in multi-domain proteins) to show RMSD of each domain, and to offer graphs showing how their relative orientations change. Regardless, an RMSD graph, by itself, will often be relatively "bland". You cannot simply look at it at conclude anything useful. That requires LOOKING at the structures: superposing snapshots along the trajectory to detect which regions are mobile, whether the individual domains remain stable, etc. From that analysis, one can often detect key distances that should be looked at more carefully: for example distances between a ligand and specific features of the protein surface, how the size of cavities varies with time, relative positions of domains, etc.

You should therefore plot (at least) the evolution of key ligand-protein distances to ensure that the ligand remains stably bound. I will send your paper to reviewers after those data are included.

---

## Round 0.2 · Major Revisions

Please address the issues highlighted by our reviewers. I will not insist in the increase of simulation length requested by reviewer#2, but all of reviewer#2's other suggestions are very important and should be closely adhered to. As a personal comment, Figure 7 (radius of gyration evolution) is irrelevant (as I already told you during the first round) and Figure 10 is very hard to read and should probably be split in several figures.

Reviewer 1 ·

Basic reporting

This manuscript uses hybrid virtual screening approaches to identify potential inhibitors of PGAM1 and the results show that these compounds can serve as good starting point to design better active selective scaffolds against PGAM1 enzyme. It can be recommended to publish after the following issues be addressed.

1. It is better to increase the simulation time for at least 100 ns to get a better insight into the interactions.
2. The descriptions in the methods section needs to be followed by clear algorithmic steps or even by a flowchart.
3. The binding free energy calculated using MM/GBSA has only the enthalpy contribution, while the total binding free energy should add the entropy contribution.
4. What method is used to analyze the hydrogen bonds?
5. The caption of Figure 10 is incomplete.

Experimental design

no comment

Validity of the findings

no comment

Reviewer 2 ·

Basic reporting

The manuscript titled “Identification of Human Phosphoglycerate Mutase 1 (PGAM1) inhibitors using hybrid virtual screening approaches” is very nicely written with technical clarity.

Experimental design

Experimental design is correct with few corrections need to be made as described below.

Validity of the findings

The logic of finding very clear and understandable for the reader, however can be validated through experimental work.

Additional comments

The work performed by Alharthy and co-workers involves addressing critical role of PGAM1 enzymes in cancer proliferation and design of its inhibitors using in silico approaches. Molecular dynamics simulation and free energy calculations were performed to explore whether structural deformation of hit compounds takes place or they remained stable through simulations. The manuscript is overall well-structured. However, its publication can be considering after addressing the following few major points.

1. 50 ns simulations are not enough to validate stability of the structure. Simulations should be extended to at least 100ns
2. Line 182 author states “The key residues involved in the interactions are Arg90 and Phe22” however Fig. illustrates that Arg90 is not key interacting residue for all five hits. (compounds). Also Tyr92 is important interacting residue for C2, C3, C4 and C5 whose effect has not be described sufficiently
3. Fig. 5, Author did not describe frame of selected trajectory neither through caption nor in the manuscript text. Also choice of allosteric site of enzyme for docking purpose needs to be elaborated.
4. Software program for Protein preparation wizard should be mentioned in line 112-113.
5. Line 141-142 of methodology section (in tracked changes manuscript) author described “production" phase at high temperature for 50 ns”. Instead, authors should clearly indicate the temperature of the system for production run. Similarly, methodology is too generalized and unclear to make a difference between NVT and NPT ensemble.
6. Line 212-214 (in tracked changes manuscript) “Five compounds with IDs C1=6144-0309, C2=E470-1348, C3=C301-8900, C4=F540-0157 and C5=L464-0403 were selected from each cluster based on glide, shape and color Tanimoto scores” need to be rephrased.
7. Figure 10 is confusing, keep the time evolution of distance separate for all of the complexes and elaborate the caption.
8. Global radius of gyration and total number of hydrogen bonds does not depict any clear picture of conformational changes associated with ligand binding. Therefore, authors should describe the conformational events taking place upon ligand binding and how these conformational changes at an allosteric site would be inhibiting the enzyme in present scenario.
9. In line 310-311 authors quoted “Glu19 and Glu89 residue showed unfavorable interactions with C2, C3, and C5 hit, as these interactions had negative contribution in total binding free energies” however these residues were untraceable in referred figure. Also negative free energy indicates that interactions are favorable and spontaneous, whereas author has described it unfavorable.
10. Instead of showing cocrystalized ligand protein interactions my suggestion would be to show a comparison of ligand protein interactions before and after MD simulation.
11. A more in-depth discussion that correlates to other previously reported inhibitors against the enzyme could be done.

Minor points
(i) Formatting –throughout manuscript writing style should be homogenous, not a mix of Arial and Time New Roman. Grammatical mistakes should be verified before publication e.g, line 247 correct spellings of insulin.
(ii) Abbreviations should be avoided the first time they appear in the abstract and introduction. e.g, in the case of AUC and ROC.

Reviewer 3 ·

Basic reporting

Clear and unambiguous

Experimental design

The experimental design is reasonable

Validity of the findings

The results are based on molecular simulation experiments, and it is recommended to supplement biological experiments to verify them

Additional comments

1. The first two paragraphs of the results and discussion are mostly theoretical descriptions, It should be placed in the Introduction
2.Arg90 and Phe22 are defined as key amino acids, but in molecular docking results, most candidate compounds do not interact with these two key amino acids
3.The Binding Free Energy Studies result units in the table are all KJ/mol, but in the results and discussion and conclusion sections, the units are Kcal/mol

---

## Round 0.3 · Minor Revisions

Please address the final comments. I will not insist on longer simulations, but I must insist on robust analysis of the stability of the complexes (for example, by showing low variability of key geometric features fo the ligand-protein interaction along the simulaiton)

Reviewer 1 ·

Basic reporting

no comment

Experimental design

no comment

Validity of the findings

no comment

Additional comments

I recommend the manuscript to be accepted for publication as it.

Reviewer 2 ·

Basic reporting

NA

Experimental design

NA

Validity of the findings

NA

Additional comments

Following comment in response to rebuttal of first review is submitted as below;

Author replied that "50 ns simulation is acceptable to follow the Journal policy. However, many studies have also shown that simulations performed till 50 ns or upto 100ns yield almost similar results".

Reviewer’s response: Instead of journal acceptable policy I am more interested to get insight in to the stability of the complex. New and better inhibitors has been reported by the authors which are expected to bind at the allosteric site to maintain the stability. These events varies from protein to protein as well as on the nature of approaching ligand. In these type of computational studies multiple simulations of a minimum of 100ns simulation is generally recommended. Other approaches include multiple short simulations and enhanced sampling approach to get an insight in to ligand binding with consistent statistics.

Reviewer 3 ·

Basic reporting

Clear and unambiguous, professional English used throughout

Experimental design

reasonable

Validity of the findings

The results of the study are reasonable

---

## Round 0.4 · accepted · Accept

I am now satisfied with the authors' replies and with the state of the manuscript.